# Cloning, Expression Analysis, and Functional Characterization of Candidate Oxalate Transporter Genes of *HbOT1* and *HbOT2* from Rubber Tree (*Hevea brasiliensis*)

**DOI:** 10.3390/cells11233793

**Published:** 2022-11-27

**Authors:** Zongming Yang, Pingjuan Zhao, Wentao Peng, Zifan Liu, Guishui Xie, Xiaowei Ma, Zewei An, Feng An

**Affiliations:** 1College of Tropical Crops, Hainan University, Haikou 570228, China; 2Hainan Danzhou Agro-Ecosystem National Observation and Research Station, Rubber Research Institute of Chinese Academy of Tropical Agricultural Sciences, Danzhou 571737, China; 3Institute of Tropical Bioscience and Biotechnology, Chinese Academy of Tropical Agricultural Sciences, Haikou 571101, China

**Keywords:** *Hevea brasiliensis*, *HbOT1*, *HbOT2*, aluminum toxicity, oxalate transporter, gene expression, functional identification

## Abstract

Secretion of oxalic acid from roots is an important aluminum detoxification mechanism for many plants such as *Hevea brasiliensis* (rubber tree). However, the underlying molecular mechanism and oxalate transporter genes in plants have not yet been reported. In this study, the oxalate transporter candidate genes *HbOT1* and *HbOT2* from the rubber tree were cloned and preliminarily identified. It was found that *HbOT1* had a full length of 1163 bp with CDS size of 792 bp, encoding 263 amino acids, and *HbOT2* had a full length of 1647 bp with a CDS region length of 840 bp, encoding 279 amino acid residues. HbOT1 and HbOT2 were both stable hydrophobic proteins with transmembrane structure and SNARE_assoc domains, possibly belonging to the SNARE_assoc subfamily proteins of the SNARE superfamily. qRT-PCR assays revealed that *HbOT1* and *HbOT2* were constitutively expressed in different tissues, with *HbOT1* highly expressed in roots, stems, barks, and latex, while *HbOT2* was highly expressed in latex. In addition, the expressions of *HbOT1* and *HbOT2* were up-regulated in response to aluminum stress, and they were inducible by metals, such as copper and manganese. Heterologous expression of *HbOT1* and *HbOT2* in the yeast mutant AD12345678 enhanced the tolerance to oxalic acid and high concentration aluminum stress, which was closely correlated with the secretion of oxalic acid. This study is the first report on oxalate transporter genes in plants, which provides a theoretical reference for the study on the molecular mechanism of oxalic acid secretion to relieve aluminum toxicity and on aluminum-tolerance genetic engineering breeding.

## 1. Introduction

Aluminum (Al) is the most abundant metal element in the earth’s crust, accounting for approximately 7.45% of the total crustal material [1]. When the soil is neutral or weakly acidic, aluminum generally exists in the form of stable silicate minerals or oxides [2]. However, when the soil pH value is less than 5, aluminum will be released from the solid phase into the soil solution and then adsorbed on the cation exchange site of the soil surface in the form of exchangeable aluminum [3]. The concentration of exchangeable aluminum in soil will resultantly be increased, leading to aluminum toxicity and adverse effects on plant growth and soil microbial activities [4]. Currently, the problem of soil acidification has become increasingly serious due to the increased frequency of global acid rain and excessive use of chemical fertilizers. The aluminum toxicity in soil has consequently become an important factor affecting global crop yields [5]. Therefore, studying the detoxification mechanism of aluminum in plants is of great practical significance [6,7].

Plants have developed two physiological mechanisms to relieve aluminum toxicity, namely external exclusion mechanism and internal detoxification mechanism [8,9,10,11]. The external exclusion mechanisms include secretion of aluminum ligands (organic acids and phosphates), immobilization of aluminum by cell wall, formation of a rhizosphere pH barrier, and efflux of Al^3+^ by consuming ATP. The internal detoxification mechanisms include complexation of aluminum with organic acids and phenols, storage of aluminum in vacuoles, formation of some proteins, and change of related enzyme activities [12]. For the external exclusion mechanism, secretion of aluminum ligands, including mainly organic acids and phosphates, is the most important pathway in plants to detoxify aluminum [13]. Phosphate plays a role in detoxification of aluminum in plant cell walls, while most of the aluminum ligands secreted into plants are organic acids [14]. Organic acids secreted in vitro will form stable complexes with aluminum, thus preventing aluminum from entering the eutectic [15,16]. For the internal tolerance mechanism, organic acid anions and phenolic compounds complex with aluminum, forming low toxic organic aluminum complexes, which is a prerequisite for plants to tolerate high concentrations of aluminum in vivo [17]. Therefore, aluminum-induced synthesis and secretion of organic acids play a very pivotal role in the aluminum tolerance mechanism of plants.

Citric acid, malic acid, and oxalic acid are the main organic acids related to the detoxification of aluminum in plants [18,19,20]. *TaALMT1*, the first malate transporter gene inducing the secretion of malate from the root tips of plants for aluminum detoxification, was identified from wheat [21]. Subsequently, genes encoding malate transporters have been found in many plants, and are considered to be related to many stress responses, such as *AtALMT1* in Arabidopsis [22], *BnALMT1*, *BnALMT2* in rape [23], *ScALMT1* in rye [24], and *GmALMT1* in soybean [25]. At present, the citrate transporter genes have also been identified in Arabidopsis [26], barley [27], sorghum [28], and other plants. The mechanism of plant roots to secrete citrate and chelate Al^3+^ has been proved. Secretion of oxalic acid by roots is an important aluminum detoxification mechanism for many plants, such as taro, buckwheat, tea, spinach, tomato, and polygonum [15,16,18]. However, except for Lv et al. (2021), who screened an aluminum-induced expression of *MsDHN1* as an oxalate secretion-related regulatory gene through transcriptome analysis of alfalfa under aluminum stress [29], the oxalate transporter in plants has not yet been identified and reported.

The rubber tree (*Hevea brasiliensis*) is an important economic tree species planted mainly in tropical and subtropical regions. In recent years, soil pH values in the southern China and Southeast Asia rubber plantations have decreased significantly, which makes the content of soil-exchangeable aluminum close to the level that is toxic to crops, posing a systemic risk for rubber plantation recession or death [5,30]. Our previous studies have shown that the aluminum tolerance range of the rubber tree is 100–200 mmol/L, indicating comparatively high tolerance to aluminum [31,32]. When rubber saplings are cultured in a high aluminum environment, large amounts of oxalic acid are secreted from root tips to relieve aluminum toxicity. Furthermore, exogenous application of oxalic acid can significantly alleviate the aluminum toxicity to rubber saplings [33], indicating that oxalic acid plays a vital role in the aluminum tolerance mechanism of rubber trees. However, the oxalate transporter gene in plants has not yet been reported, and the molecular mechanism of secreting oxalate from the roots of rubber trees to relieve aluminum toxicity remains unclear.

In fungi, the efflux of oxalic acid is an important mechanism of wood decay caused by wood-rot fungi, such as brown rot fungus and white rot fungus. Therefore, the biochemical role of oxalic acid in wood-rot fungi has been attracting much attention. Watanabe et al. studied the mechanism of oxalic acid secretion by brown rot fungus (*Fomitopsis palustris*) in the catalytic process of wood decay by using the yeast mutant AD12345678 (AD1-8) that lacks the transporter function and identified an oxalate transporter gene *FpOAR* (*F. palustris* oxalic acid resistance), which was capable of transporting intracellular oxalic acid to the outside by consuming ATP in brown rot fungus [34]. On the basis of this discovery, we performed a BLASTP homology alignment analysis with FpOAR, and screened two highly homologous Unigene sequences in the transcriptome and genome database of rubber tree roots, tentatively named *HbOT1* and *HbOT2* (*H. brasiliensis* oxalate transporter). In this study, bioinformatics analysis, expression analysis, and yeast genetic transformation system were used to examine whether *HbOT1* and *HbOT2* were aluminum-responsive oxalate transporter genes in rubber trees.

## 2. Materials and Methods

### 2.1. Plant Materials

The rubber tree saplings of “Reyan 7-33-97” variety with two-whorled leaves were selected as plant materials. They were hydroponically cultured in the Hoagland nutrient solution (2.8 mg/L H_3_BO_3_, 3.4 mg/L MnSO_4_·H_2_O, 0.1 mg/L CuSO_4_·5H_2_O, 0.22 mg/L ZnSO_4_·7H_2_O, 0.1 mg/L (NH_4_)_6_Mo_7_O_24_·4H_2_O, 20 mg/L Na_2_Fe-EDTA, 0.94 g/L Ca(NO_3_)_2_·4H_2_O, 0.52 g/L MgSO_4_·7H_2_O, 0.66 g/L KNO_3_, and 0.12 g/L NH_4_H_2_PO_4_) in a growth chamber under the irradiation intensity of 200 μmol∙m^−2^∙s^−1^ at 28 °C for 16 h and then in the dark at 25 °C for 8 h. The rubber tree saplings were grown for 120 h in Hoagland nutrient solution (containing 40 μmol/L AlCl_3_, pH = 5.5) before aluminum stress treatments to prevent the possible shock response of aluminum stress on rubber saplings [35]. The recovered rubber tree saplings were then treated with Hoagland solutions containing 0 (CK), 50, 100, and 200 mmol/L of AlCl_3_, respectively. During the aluminum stress treatment (including the control), the pH of Hoagland nutrient solution was adjusted to 4.2 with 1 mmol/L HCl or ammonia and updated every 2 days.

The roots, stems, leaves, and barks of the rubber tree saplings under normal conditions were collected for tissue expression analysis as control. To reveal the effect of different metal ions on relative gene expressions, roots of rubber tree saplings treated with 200 mmol/L AlCl_3_·6H_2_O, CuSO_4_·5H_2_O, PbCl_2_, CdCl_2_, MnSO_4_·H_2_O, and LaCl_3_·7H_2_O for 120 h were collected for gene expression analysis. To investigate the effect of Al stress concentration on *HbOT1* and *HbOT2* expressions, roots of rubber tree saplings treated with different concentrations of Al^3+^ for 120 h were collected for gene expression analysis. To analyze the effect of Al stress on gene expressions, the roots of rubber tree saplings treated with 200 mmol/L Al^3+^ for 0, 6, 12, 24, 48, and 120 h were collected for gene expression analysis. All samples were frozen in liquid nitrogen and stored at −80 °C.

### 2.2. RNA Extraction, cDNA Synthesis, and qRT-PCR Analysis

According to the manual of TIANGEN plant total RNA extraction kit (Tiangen Biochemical Technology Co., Ltd., Beijing, China), total RNA was extracted from different tissues and treated samples of rubber trees. Then, the concentration and purity of RNA were analyzed by a Thermo Fisher NanoDrop 2000 ultra-micro nucleic acid protein analyzer (Thermo Fisher Technology (China) Co., Ltd., Shanghai, China). Subsequently, the RNA extracted was reversely transcribed into cDNA according to the manufacturer’s instructions of the TaKaRa reverse transcription kit (Baori Doctor Technology Co., Ltd., Dalian, China).

The qRT-PCR analysis was conducted using a CFX96 TOUCH real-time fluorescent quantitative PCR instrument (Bio-Rad, Hercules, CA, USA), and primers used for qRT-PCR are listed in Table 1. The *HbUBC4* of the rubber tree with the most stable expression under aluminum stress was selected as the internal reference gene [36]. The PCR reaction mixture was 20 μL in volume, which consisted of SYBR Premix Ex TaqTM (TaKaRa) 10 μL, each upstream and downstream primers (10 μmol/μL) 0.4 μL, cDNA 1 μL, and ddH_2_O 8.2 μL. The reaction procedure, including the first step, was conducted at 95 °C for 3 min and at 95 °C for 10 s, the second step at 60 °C for 20 s, and the third step at 72 °C for 30 s, for a total of 43 cycles. The test results were analyzed by the 2^−ΔΔCT^ method.

### 2.3. Cloning of HbOT1 and HbOT2 and Sequence Analysis

Specific primers for the ORF sequences of *HbOT1* and *HbOT2* were designed according to the NCBI rubber tree database (Table 1). PCR technology was used to amplify the ORF sequences of *HbOT1* and *HbOT2* with the root cDNA of “Reyan 7-33-97” rubber tree saplings as templates. By referring to the OMEGA gel recovery kit (Feiyang Bioengineering Co., Ltd., Guangzhou, China) instructions, the target gene fragments were purified, cloned into the pMD-18T vector, and, finally, sequenced.

Bioinformatics analysis of HbOT1 and HbOT2 was performed using online tools (Appendix A). DNAMAN 7 software was used for multi-sequence alignment and homologous similarity analysis, and MEGA 7 software was used for phylogenetic analysis. The protein sequences selected for phylogenetic analysis were retrieved from NCBI database.

To confirm the subcellular localization of HbOT1 and HbOT2, the coding regions of *HbOT1* and *HbOT2* were inserted into the pCAMBIA1302 and pCAMBIA1300 vectors using specific primers (Table 1) containing the CaMV 35S promoter and a GFP gene, generating the *35S-HbOT1-GFP* and *35S-HbOT2-GFP* genes, respectively. The constructed *35S::HbOT1-GFP* and *35S::HbOT2-GFP* vector were introduced into the *Agrobacterium tumefaciens* strain GV3101 by the heat shock method. Transformants were selected using kanamycin (50 μg/mL) and then suspended in MMA buffer (10 mmol/L MgCl_2_, 10 mmol/L MES and 100 μmol/L acetosyringone). The *A. tumefaciens* suspensions were subsequently injected into leaves of 4-week-old *Nicotiana benthamiana* plants and cultivated at 25 °C for 24h in the dark. After 48h of infiltration, the transient expressions of HbOT1 and HbOT2 were detected using a LSM800 confocal laser scanning microscope (Carl Zeiss Shanghai Co. Ltd., Shanghai, China).

### 2.4. Yeast Transformation and Stress Tolerance Assays

Yeast mutant AD1-8 (donated by Professor Tang of Shanghai Jiaotong University and Professor Richard Cannon of Otago University) and pDR196 vector (Beijing Zhuang Meng International Biogene Technology Co., Ltd., Beijing, China) were used to further identify the oxalate transporter and aluminum tolerance function of *HbOT1* and *HbOT2*, respectively. *Sal* I and *EcoR* I were selected as restriction sites to construct recombinant expression vectors pDR196-*HbOT1*, pDR196-*HbOT2*, and pDR196-*FpOAR* (positive control) by homologous recombination with a ClonExpress^®^ II One Step Cloning Kit (Vazyme Biotech Co., Ltd., Nanjing, China). The recombinant expression vector and pDR196 empty vector (negative control) were transformed into yeast mutant AD1-8 by lithium acetate transformation. The yeast transformants were cultured in SD (-Ura) (Aili Biotechnology Co., Ltd., Shanghai) liquid on a shaker at 30 °C with the rotate speed of 180 rpm until OD_600_ = 0.5. Then, they were diluted with ddH_2_O to 10^−1^, 10^−2^, 10^−3^, 10^−4^, and 10^−5^ and cultivated on SD (-Ura) plates containing 0 (CK), 2, 4, 8, and 10 mmol/L oxalic acid separately for 4 days at 30 °C in order to record their growth status [34]. The growth status of yeast cells in the environment containing 2.4, 2.5, 2.6, 2.7, and 2.8 mmol/L Al^3+^ was recorded using the same method as above.

A total of 100 μL of each yeast cell solution with OD_600_ = 1.0 was cultured with 50 mL of SD (-Ura) containing 2 mmol/L oxalic acid culture on a shaker at 30 °C under the rotate speed of 180 rpm for 0, 1, 3, 5, 7, 9, 11, and 13 days. The bacteria solution was centrifuged at 1000× *g* for 10 min to separate the yeast cells from the medium for the determination of oxalic acid content using boxbio oxalic acid (OA) content determination kit (Box Biotechnology Co., Ltd., Beijing, China). METTLER TOLEDO FE plus pH meter (Metler Toledo Technology Co., Ltd., Shanghai, China) was used to determine the pH value of bacteria solution under different culture times. Yeast cells were freeze-dried by SCIENTZ-30YG/A freeze-drying machine (Xinzhi Freeze-drying Equipment Co., Ltd., Ningbo, China) before measuring their dry weight. The protein carbonyl content in yeast cells under aluminum stress was determined by DNPH method, as described in [37].

### 2.5. Statistical Methods

The gene expression level was given as mean ± standard deviation from 3 biological and 3 technical repetitions. The pH value, dry cell weight, and oxalic acid content of yeast solution were expressed as mean ± standard deviation of 3 biological repetitions. Microsoft Office Excel was used for data analysis and mapping. Single-factor ANOVA test was performed to analyze the significance of difference in the IBM SPSS Statistics 25 software following the Duncan’s new multiple range method.

## 3. Results

### 3.1. Cloning of HbOT1 and HbOT2 from Rubber Tree

With the identified FpOAR protein (GeneBank: BAJ10704.1) of *F. palustris* as the query sequence, two unidentified rubber tree proteins containing the SNARE_assoc superfamily conserve domain, XP_021645179.1 and XP_021655511.1, were screened by BLASTP alignment, and their corresponding mRNAs, XM_021789487.1 and XM_021799819.1, were obtained. Meanwhile, a SNARE-associated Golgi protein, NP_192696.3, that is homologous to HbOT1 and HbOT2 in *Arabidopsis thaliana* (AtOT), was used to predict their functions (Appendix A). Since the function of AtOT is also not clear, the ORF sequences of the two rubber tree target genes were obtained by PCR (Figure 1) and temporarily named *HbOT1* (ID_NCBI: LOC110638805) and *HbOT2* (ID_NCBI: LOC110646400), respectively. *HbOT1* had a length of 1163 bp, with the CDS region size of 792 bp, encoding 263 amino acid residues, while *HbOT2* had a length of 1647 bp, with the CDS region size of 840 bp, encoding 279 amino acid residues.

### 3.2. Characterization of HbOT1 and HbOT2

The basic physicochemical properties of the two proteins were analyzed on the ProtParam online website. It was found that HbOT1 had the relative molecular weight 28,630.58 Da, the theoretical isoelectric point 9.61, the instability index 33.70, and the total average hydrophilicity 0.473, while the HbOT2 had the relative molecular weight 31,448.44 Da, the theoretical isoelectric point 9.71, the instability index 37.40, and the total average hydrophilicity 0.411. In addition, the TMPRED online website was used to predict the transmembrane domain. The results showed that there were four transmembrane helixes in HbOT1, including 46–68, 78–100, 159–181, and 201–220, and five transmembrane helixes in the amino acid sequence of HbOT2, including 45–67, 103–125, 135–157, 218–240, and 255–272. Both HbOT1 and HbOT2 were transmembrane proteins. The online website SignalIP was used to predict the signal peptide. The results showed that the possibilities of HbOT1 and HbOT2 being signal peptides were 0.0376 and 0.0003, respectively, which were less than the threshold of 0.5000. Amino acid hydrophilic/hydrophobic analysis was carried out on the basis of ProtScale online website, and the results showed that the hydrophobic parts of HbOT1 and HbOT2 were greater than the hydrophilic parts.

The conserved domains of protein were analyzed by using the NCBI CDD database and SMART online website. The results showed that HbOT1 had a SNARE_assoc domain, which was located in the 62~184 N-terminal, and the E-value was 4.88 × 10^−20^. Similarly, HbOT2 had a SNARE_assoc domain, which was located in the 85~278 N-terminal, and the E-value was 1.51 × 10^−17^ (Figure 2A). The secondary structure of the protein was predicted using the online website NPS @ SOPMA (Figure 2B). The results showed that there were 126, 41, 19, and 77 amino acid residues involved in the formation of the α-helix, outer extension chain, β rotation angle, and random coil in the amino acid sequence-encoding HbOT1 protein, accounting for 47.91%, 15.59%, 7.22%, and 29.28% of the secondary structure, respectively. While the amino acid sequence-encoding HbOT2 protein contained 156, 32, 13, and 78 amino acid residues involved in the α-helix, extended strand, β-turn, and random coil, accounting for 55.91%, 11.47%, 4.66%, and 27.96% of the secondary structure, respectively. The tertiary structure models of HbOT1 and HbOT2 constructed by I-TASSER are shown in Figure 2C, which is basically consistent with the secondary structure prediction results.

The subcellular localizations of HbOT1 and HbOT2 were predicted on the basis of the online website PSORT. The prediction showed that HbOT1 and HbOT2 might be located on the plasma membrane and vacuole, with confidence scores of 6 and 5 for HbOT1 and 10 and 2 for HbOT2, respectively. Tobacco leaf epidermal cells containing the *35S::HbOT1-GFP* and *35S::HbOT2-GFP* fusion protein were used to verify the subcellular localization, and it was found that fluorescence could be in the plasma membrane only, whereas fluorescence was found throughout the cell for the *35S::1300-GFP* fusion protein injection (Figure 3). These results implied that HbOT1 and HbOT2 were localized on the plasma membrane rather than on the vacuole.

### 3.3. Multiple Sequence Alignment and Phylogenetic Tree Analysis of HbOT1 and HbOT2

The amino acid sequences of HbOT1 and HbOT2 were searched by NCBI BLASTP. The results showed that they had high homology with SNARE proteins of *Arabidopsis thaliana* (AtOT, NP_175116.2), cassava (*Manihot esculenta*, XP_021601880.1, XP_021621868.1), castor (*Ricinus communis*, XP_015579430.1), *Jatropha curcas* (XP_012065770.1, KDP23232.1), *Prunus mume* (XP_008237264.1), *Prunus persica* (XP_007221446.1), *Populus deltoides* (KAH8513031.1), sweet cherry (*Prunus avium*, XP_021834470.1), *Populus trichocarpa* (XP_002325990.2, XP_006376293.1), *Populus alba* (XP_034924621.1, XP_034897016.1), and *Juglans regia* (XP_018829221.1). HbOT1 had the highest homology with cassava SNARE protein XP_021621868.1, which was 90.87%, and HbOT2 had the highest homology with cassava SNARE protein XP_021601880.1, which was 96.42%. On the basis of the assumption that HbOT1 and HbOT2 proteins of rubber tree were localized in the plasma membrane, the SNARE protein subfamilies of other membrane-localized plants were introduced for comparative analysis. The phylogenetic tree was constructed by MEGA 7 software and the neighbor-joining method was used to further explore the evolutionary relationship among HbOT1, HbOT2, and SNARE proteins of other plants. The results showed that HbOT1 and HbOT2 belong to the SNARE_ assoc subfamily, and the common feature of the subfamily proteins is that they have the SNARE_ assoc domain. In the evolutionary relationship, HbOT1 and HbOT2 were located closest to the cassava SNARE proteins XP_021621868.1 and XP_021601880.1, respectively, indicating their functional similarities (Figure 4).

### 3.4. Tissue-Specific Expression of HbOT1 and HbOT2

Tissue-specific expression of *HbOT1* and *HbOT2* was analyzed by qRT-PCR (Figure 5A,B). The results showed that *HbOT1* and *HbOT2* were expressed in roots, stems, leaves, bark, and latex of rubber tree. The relative expression of *HbOT1* was significantly higher in root, stem tip, bark, and latex than that in leaf, and there was no significant difference in relative expression of *HbOT1* in root, stem tip, bark, and latex. The relative expression of *HbOT2* was the highest in latex, which was significantly higher than that in root, stem tip, leaf, and bark, and the relative expressed was the lowest in bark, root, and stem tip.

### 3.5. The Expression Pattern of HbOT1 and HbOT2 in Response to Various Metal Ion Stresses

Excessive metal elements in soil can adversely affect the growth of plants. Among them, aluminum (Al), copper (Cu), lead (Pb), cadmium (Cd), manganese (Mn), and lanthanum (La) stresses have been widely reported as research hotspots. The rubber tree saplings were treated with 200 mmol/L Al^3+^, Cu^2+^, Pb^2+^, Cd^2+^, Mn^2+^, and La^3+^ for 120 h, respectively (Figure 5C,D). It was found that the gene expression of *HbOT1* was significantly up-regulated under Al and Cu stress, with the expression levels being 2.92 times and 2.86 times that of the CK level, respectively, but there was no significant change in gene expression under Cd, Pb, Mn, or La stress. In contrast, the gene expression of *HbOT2* was significantly up-regulated under Al, Cd, and Mn stress, which were 5.86, 3.46, and 12.58 times that of the CK level, respectively, but there was no significant change in gene expression under Cu, Pb, or La stress.

### 3.6. The Expression Pattern of HbOT1 and HbOT2 in Response to Aluminum Stresses

The rubber tree saplings were treated with different concentrations of aluminum stress. It was found that the expressions of *HbOT1* and *HbOT2* were significantly up-regulated under 50 mmol/L, 100 mmol/L, and 200 mmol/L of Al^3+^ compared with CK after 120 h of aluminum stress. Under 50, 100, and 200 mmol/L of Al^3+^ stresses, the expression of *HbOT1* was 1.78, 1.64, and 2.40 times that of the CK level, and the expression of *HbOT2* was 15.34, 2.91, and 4.89 times that of the CK level, respectively (Figure 5E,F).

The expressions of *HbOT1* and *HbOT2* under 200 mmol/L Al^3+^ stress for different treatment time were further studied. It was found that the gene expression of *HbOT1* was significantly up-regulated at 6 h, 12 h, and 24 h, with the expression levels being 346.87, 99.59, and 695.32 times that of the CK level, respectively, while the gene expression changed insignificantly 48 h and 120 h after the Al^3+^ treatments. In contrast, the gene expression of *HbOT2* was significantly up-regulated at 6 h, 12 h, 24 h, 48 h, and 120 h, which were 9.83, 10.66, 11.38, 5.41, and 4.50 times that of the CK level, respectively (Figure 5G,H).

### 3.7. Oxalic Acid Resistance and Oxalate Transporter Function Identification of HbOT1 and HbOT2 in Yeast

To elucidate the role for *HbOT1* and *HbOT2* in response to oxalic acid, we heterologously overexpressed *HbOT1* and *HbOT2* in yeast mutant AD1-8. The transgenic yeast cells with *FpOAR* were used as the positive control, and the transgenic yeast cells with pDR196 empty vector were used as the negative control. Yeast cells with different concentration gradients were cultured in SD (-Ura) plates containing 0, 2, 4, 8, and 10 mmol/L oxalic acid for 4 days, respectively. The results showed that when the oxalic acid concentration was not higher than 2 mmol/L, the growth status of all yeast cells was similar, except that three white needle-like colonies were formed by the negative control at 2 mmol/L oxalic acid at yeast concentration of 10^−5^, indicating that the yeast mutant AD1–8 strain itself had certain resistance to low concentration (≤ 2 mmol/L) of oxalic acid stress. When the oxalic acid concentration reached 4–8 mmol/L, the negative control had no colonies at yeast concentrations of 10^−3^, 10^−4^ and 10^−5^, and the colonies formed by the positive control and *HbOT1*- and *HbOT2*-transformed yeasts changed from white smooth round colonies to white waveform dot colonies with the decrease of yeast cell concentrations. When the oxalic acid concentration reached 10 mmol/L, at yeast concentration of 10^−1^, the positive control formed three white small dot colonies, which were too tiny to be discovered, *HbOT1*- and *HbOT2*-transformed yeasts formed white irregular colonies, and the negative control had no colony formation. No colony formation was observed at yeast concentrations of 10^−2^, 10^−3^, 10^−4^, and 10^−5^ when the oxalic acid concentration reached 10 mmol/L (Figure 6). These results suggest that *HbOT1* and *HbOT2* have a positive regulatory effect on oxalic acid resistance of yeast cells in colony phenotype, and the oxalic acid resistance ability is stronger than that of the positive control *FpOAR*. In addition, it was found that the growth of negative control yeast cells was inhibited in the liquid environment when the concentration of oxalic acid reached 2 mmol/L, indicating that 2 mmol/L was the maximum concentration at which oxalic acid may exert stress on the yeast strain in the liquid culture conditions.

Different yeast cells were cultured in SD (-Ura) liquid medium containing 2 mmol/L oxalic acid for 13 days and then continuously observed every 48 h. It was found that the pH of each yeast solution showed a downward trend with the passage of culture time. Except that the pH of the positive control on the 11th day was significantly higher than that of *HbOT1*- and *HbOT2*-transformed yeasts, the pH value and pH variation trend of *HbOT1*- and *HbOT2*-transformed yeasts and the positive control were similar, in which the pH value continuously declined, and the negative control was always significantly higher than that of *HbOT1*- and *HbOT2*-transformed yeasts and the positive control. At the end of the culture, the pH value of each yeast reached the critical value, with the pH value of the *HbOT1*- and *HbOT2*-transformed yeasts and positive control maintained at 2.3–2.5, and that of the negative control maintained at approximately 2.8 (Table 2).

The dry weight of each yeast cell showed an upward trend with extension of the culture time. The dry weight of the negative control was significantly lower than that of *HbOT1*- and *HbOT2*-transformed yeasts and the positive control, which was maintained at approximately 0.2 g at the end of culture. There were little differences in dry weight and dry weight trend among *HbOT1*- and *HbOT2*-transformed yeasts. The dry weight variations of *HbOT1*- and *HbOT2*-transformed yeasts and the positive control were not significantly different from the 0 to 7th day. However, the dry weight of the positive control, which was maintained at 0.225–0.25 g finally, was significantly lower than that of *HbOT1*- and *HbOT2*-transformed yeasts from the 7th to 13th day. Meanwhile, the dry weight of *HbOT1*- and *HbOT2*-transformed yeasts reached 0.325 g at the end of culture, which was the highest (Table 3).

The oxalic acid content in yeast was determined with an oxalic acid (OA) content determination kit by the sulfosalicylic acid method (Figure 7A). The results showed that the oxalic acid content in yeast cells increased significantly from 3.5–5 mmol/L to approximately 10 mmol/L in a short time (≤1 day) under the oxalic acid environment, and there was no significant difference in the oxalic acid content among different recombinant yeast cells. From the 3rd day and later, however, the oxalic acid contents of *HbOT1*- and *HbOT2*-transformed yeasts and positive control yeast decreased with the passage of culture time, which was maintained at approximately 3–4.5 mmol/L at the end of culture. The oxalic acid content of negative control yeast was always at a high level at all culture times. At the end of culture, the oxalic acid content of negative control yeast was maintained at 9.02–11.20 mmol/L, which was significantly higher than that of *HbOT1*- and *HbOT2*-transformed yeasts and positive control.

The oxalic acid content in the medium was also determined to verify whether the transformed yeast can transport oxalic acid outward (Figure 7B). It was found that the oxalic acid content in the medium of negative control was not stable with the passage of culture time, but it was significantly lower than that in *HbOT1*- and *HbOT2*-transformed yeasts and positive control from the 7th day to the end of culture. With the extension of culture time, oxalic acid content in the medium of *HbOT1*- and *HbOT2*-transformed yeasts and positive control showed an overall upward trend. The oxalic acid content in the medium of positive control increased to the critical value on the 7th day, then fluctuated at approximately 8.5 mmol/L until the end of culture. However, the oxalic acid content in the medium of *HbOT1*- and *HbOT2*-transformed yeasts increased gradually, and the oxalic acid content in the medium of the yeasts transferred with *HbOT1* and *HbOT2* reached 14.17 mmol/L and 12.04 mmol/L at the end of culture, respectively. These results indicated that *HbOT1*- and *HbOT2*-transformed yeasts had the ability to transport oxalic acid to the medium.

The above experimental results showed that oxalic acid stress affected the normal growth of yeast cells and disturbed the oxalic acid metabolism of the negative control. *HbOT1* and *HbOT2* changed the oxalic acid metabolism of the *HbOT1*- and *HbOT2*-recombinant yeast cells in oxalic acid environment and played a role in the efflux of oxalic acid from inner cells, which significantly enhanced the oxalic acid adaptability of the *HbOT1*- and *HbOT2*-transformed yeast cells.

### 3.8. Identification of Aluminum Tolerance Function of HbOT1 and HbOT2 in Yeast

In order to further verify the role of *HbOT1* and *HbOT2* in yeast aluminum tolerance, yeast transformants were cultured in different concentrations of aluminum stress for 4 days (Figure 8). The results showed that the yeast transformed with *HbOT1* and *HbOT2* showed stronger aluminum tolerance than the negative control. When the concentration of Al^3+^ reached 2.4 mmol/L, the colony size of each yeast cell was significantly smaller than that under normal conditions, and there was no significant difference among them. When the concentration of Al^3+^ was 2.5–2.6 mmol/L, there was no significant difference in the growth of yeast cells at the 10^−1^ yeast concentration, and no colonies were formed at the concentrations of 10^−4^ and 10^−5^. When the concentration of Al^3+^ reached 2.7 mmol/L, the yeasts transformed with *FpOAR* and *HbOT1* developed white smooth round colonies at yeast concentration of 10^−1^, the yeasts transformed with *HbOT2* formed white dot colonies, and the negative control yeast could not grow. At the concentrations of 10^−2^ and 10^−3^, the yeasts transformed with *FpOAR* and *HbOT2* formed white needle-like colonies, and the yeasts transformed with *HbOT1* and blank control showed no colony formation. At the concentrations of 10^−4^ and 10^−5^, no colonies were observed in any yeast cells. When the concentration of Al^3+^ reached 2.8 mmol/L, the yeasts transformed with *HbOT1* and *HbOT2* formed white acicular colonies at the yeast concentration of 10^−1^, and no colonies were found under other dilution concentrations.

Different yeast cells were cultured in liquid medium containing 2.7 mmol/L Al^3+^ for 48 h. The OD_600_ values of bacteria solution were measured every 2 h and cell growth curves were drawn (Figure 9). It was found that the growth of each yeast cell conformed to the S-type growth curve. The *FpOAR*-transformed yeast showed a similar growth pattern to that of the negative control yeast, while the *HbOT1*- and *HbOT2*-transformed yeasts had a similar growth pattern. The *FpOAR*-transformed yeast and the negative control yeast entered the logarithmic growth phase at 36 h and reached the plateau phase at 44 h, with their OD_600_ values stabilized at approximately 2.10 and 2.24, respectively. However, *HbOT1*- and *HbOT2*-transformed yeasts entered the logarithmic growth phase at 8 h and reached the plateau phase at 22 h, with the stabilized OD_600_ values of 2.48 and 2.44, respectively.

Carbonylation of proteins is widely used to evaluate the oxidation degree of various biological organisms, and the protein carbonyl (PC) content is a sensitive indicator of protein oxidation [38]. The results showed that with the passage of aluminum treatment time, the PC content in each yeast cell showed an upward trend (Figure 10). The PC contents in *FpOAR*-transformed yeast and negative control yeast were similar, reaching 0.0121 and 0.0108 mol/g at the end of culture, respectively. Meanwhile, the PC content in *HbOT1*- and *HbOT2*-transformed yeasts was significantly lower than that in *FpOAR*-transformation yeasts and negative control yeasts, reaching 0.0056 and 0.0074 mol/g at the end of culture, respectively. This indicated that *HbOT1*- and *HbOT2*-transformed yeasts were less oxidized under aluminum stress, so they had stronger tolerance to aluminum stress.

The above results showed that *HbOT1* and *HbOT2* could improve the tolerance limit of yeast cells to aluminum stress, and positively regulate the tolerance of yeast cells to high concentrations (≥2.7 mmol/L) of aluminum stress, indicating the role of *HbOT1* and *HbOT2* in improving the aluminum tolerance of rubber trees.

## 4. Discussion

In response to the increased aluminum concentration, the leaf, root, and even the stem of rubber saplings present some toxicity symptoms [31,32]. Organic acids secretion is proved to be the most important mechanism for many plants to cope with aluminum stress [8,13]. Our previous studies [33] showed that oxalic acid was the most main organic acid secreted by roots and played the most important role in aluminum detoxification of rubber trees. However, the transporters of oxalate in plants have not been identified, and the molecular mechanism of detoxifying aluminum by oxalic acid in roots has not been fully understood.

In this study, the BLAST homology analysis method was used to analyze and clone the oxalate transporter candidate genes *HbOT1* and *HbOT2* that were homologous to oxalate transporter gene *FpOAR* of *F. palustris* from the rubber tree. The prediction analysis of conserved domains showed that HbOT1 and HbOT2 proteins had SNARE_assoc conserved domains, belonging to the SNARE superfamily.

SNARE protein plays a role in the transport and membrane fusion mechanism of the endomembrane system that has the functions of regulating vesicle synthesis, directional transport, and recognizing and promoting the fusion between vesicles and specific target membranes [39]. At present, the research on plant SNARE proteins is still in its infancy. Although a large number of SNARE homologous genes have been found in plants through genome sequence alignment, few of them have been biologically verified [40,41]. Transmembrane domain prediction analysis showed that HbOT1 and HbOT2 proteins had four and five transmembrane domains, respectively. Subcellular localization proved that HbOT1 and HbOT2 were more likely to be on the plasma membrane, and the hydrophobic part of protein was greater than the hydrophilic part. It indicated that HbOT1 and HbOT2 were hydrophobic membrane proteins with transmembrane structure. Except for the phylogenetic study of the SYP1 branch in Qa-SNAREs by Slane et al. [42], no studies on the phylogenetic evolution of the entire SNARE protein family have been reported. As an important member of the SNARE family and the Qa-SNARE subfamily, SYP121 is located in the cytoplasm membrane in most plants, participating in plant defense responses to biotic and abiotic stresses [43]. SYP122, as the homologous protein of SYP121, has a relatively close evolutionary relationship and similar function with SYP121. They are jointly involved in the negative regulation of JA, ethylene, and salicylic acid-dependent defense mechanisms [44]. Zhang et al. reported that *AtSYP121*, together with *AtSYP122*, functioned as a negative regulator of subsequently induced defense pathways [45]. Liu et al. proved the positive role of *TaSYP71* in wheat resistance against *Puccinia striiformis* f. sp. *tritici* by the yeast heterologous expression system and the virus-induced gene silencing system [46]. VAMP, as a member of the R-SNARE subfamily, is located mainly on the vesicle membrane, playing an important role in regulating the integration of vesicles and vacuole membrane and resisting osmotic stress, ion stress, and drought stress [47,48]. Gu et al. found that Arabidopsis *AtVAMP714* played a regulation role in the exocytosis of PIN-related vesicles from the Golgi body to the plasma membrane and in the circulation of PIN protein between the plasma membrane and the inner body [49]. Xue et al. found that Arabidopsis *AtVAMP711,* which is induced by ABA, interacted with AHA1/AHA2 under drought stress to inhibit plasma membrane H^+^-ATPase activity, thereby regulating stomatal closure, reducing water loss in plants, and improving plant drought tolerance [50]. Sugano et al. found that the correct positioning of rice *OsVAMP714* in chloroplast was of great importance for rice to resist rice blast disease [51]. It indicates that plant SNARE protein not only participates in the transport of endomembrane system but also regulates plant growth by interacting with ion channel proteins, playing an important role in plant resistance to pathogenic bacteria and various abiotic stresses [52,53]. After introducing the above membrane-localized SNARE subfamily proteins into the phylogenetic tree analysis, it was found that HbOT1 and HbOT2 proteins were categorized into a new subfamily. On the basis of the high conservation of its members in the evolutionary relationship and the conservative SNARE_assoc domain, we named them SNARE_assoc subfamily. However, there have been no systematic study on the localization and function of SNARE_assoc subfamily in plants. HbOT1 and HbOT2 proteins have similar structures to the SNARE family and basic properties of transporter proteins, which are likely to be involved in the transport of oxalate in cells. Therefore, HbOT1 and HbOT2 may play an important role in the transport of rubber tree vesicles and in the physiological mechanism of stress resistance.

qRT-PCR was used to analyze the expression patterns of *HbOT1* and *HbOT2*, and the results showed that *HbOT1* and *HbOT2* were expressed in different tissues. The expression of *HbOT1* was higher in root, stem tip, bark and latex, and the expression of *HbOT2* was the highest in latex, indicating that the expressions of *HbOT1* and *HbOT2* in rubber trees occur in a tissue-specific way. Furthermore, the expression of *HbOT1* was significantly up-regulated under Al and Cu stress, while the expression of *HbOT2* was significantly up-regulated under Al, Mn, and Cd stress, indicating that *HbOT1* and *HbOT2* can respond to some metal ion stresses and play a role in plants’ defense mechanism against metal ion stresses. Further research showed that Al stress could up-regulate the expressions of *HbOT1* and *HbOT2* significantly, which was consistent with the conclusion of the metal ion stress experiment, suggesting that *HbOT1* and *HbOT2* may be important regulatory factors participating in the response of rubber trees to aluminum stress.

Yeast mutant AD1-8 (Δ*yor1*, Δ*snq2*, Δ*pdr5*, Δ*pdr10*, Δ*pdr11*, Δ*ycf1*, Δ*pdr3*, and Δ*pdr*) is a yeast strain with seven transporters and one transcription factor regulating the expression of transporter genes that are knocked out [54]. The oxalic acid resistance and aluminum tolerance of *HbOT1* and *HbOT2* in the rubber tree were identified primarily by this yeast system. Through the comparison of colony phenotype, pH value, cell dry weight, and oxalic acid content in vitro and in vivo, we found that the activity of *HbOT1*- and *HbOT2*-transformed yeasts and *FpOAR* positive control were significantly stronger than that of negative control in the oxalic acid environment, which was likely related to the transport of oxalate. Moreover, it was worth noting that the oxalic acid resistance of *HbOT1* and *HbOT2* under high concentration of oxalic acid was stronger than that of the identified oxalate transporter gene *FpOAR* in *F. palustris*. Under high concentrations of aluminum stress (≥2.7 mmol/L), the empty vector pDR196-transformed yeasts, as blank control, were unable to grow, while the yeast transformed with *HbOT1* and *HbOT2* could still form white colonies visible to the naked eye. Combined with the growth curve of yeast cells under aluminum stress and the change of PC content in vivo, it was indicated that HbOT1 and HbOT2 increased the aluminum tolerance limit of yeast cells and promoted the tolerance of yeast cells to high aluminum stress. However, our study observed only the change of oxalic acid content of yeast in vivo and in vitro, and preliminarily proved the possibility of oxalic acid efflux by *HbOT1* and *HbOT2*. However, whether *HbOT1, HbOT2*, and *FpOAR* share the same molecular mechanism in enhancing oxalic acid resistance by inducing oxalic acid efflux in vivo to achieve tolerance to high concentrations of oxalic acid environment remains unclear. On the other hand, *HbOT1* and *HbOT2* showed stronger aluminum tolerance only under high concentrations of aluminum stress; thus, further study is needed to investigate whether and how *HbOT1* and *HbOT2* positively regulate the efflux of oxalic acid and then chelate with Al^3+^ to alleviate the harm of aluminum toxicity to rubber trees. In addition, phylogenetic tree analysis showed that HbOT1 and HbOT2 proteins were consistent with the SNARE_assoc subfamily of other SNARE families in terms of evolutionary relationship and structure. Whether the genes of the family have similar functions is also worthy of further discussion.

Two candidate oxalate transporter genes, namely *HbOT1* and *HbOT2*, were cloned and identified from *H. brasiliensis*, which were similar to the known oxalate transporter gene *FpOAR* of *F. palustris*. *HbOT1* and *HbOT2* were up-regulated in response to aluminum stress and were inducible by metals, such as copper and manganese. It was found that both *HbOT1* and *HbOT2* had oxalic acid resistance and oxalate transport function for the transformed yeast systems. Through comparing the colony phenotype and the oxidation of *HbOT1*- and *HbOT2*-transformed yeast under aluminum stress, it was proved that *HbOT1*- and *HbOT2*-transformed yeast had enhanced tolerance to high concentrations of aluminum. Moreover, the proteins encoded by *HbOT1* and *HbOT2* belong to the SNARE superfamily, and their expression can be induced by aluminum stress. Therefore, it is speculated that *HbOT1* and *HbOT2* play an important role in regulating the membrane vesicle transport related to oxalate and in the mechanism of secreting oxalic acid to detoxify aluminum under aluminum stress in rubber trees. This study is the first report on oxalate transporters in plants, which provides a theoretical reference for the study on molecular mechanism of oxalate secretion and aluminum-tolerance genetic engineering breeding in plants such as the rubber tree.

## Figures and Tables

**Figure 1 cells-11-03793-f001:**
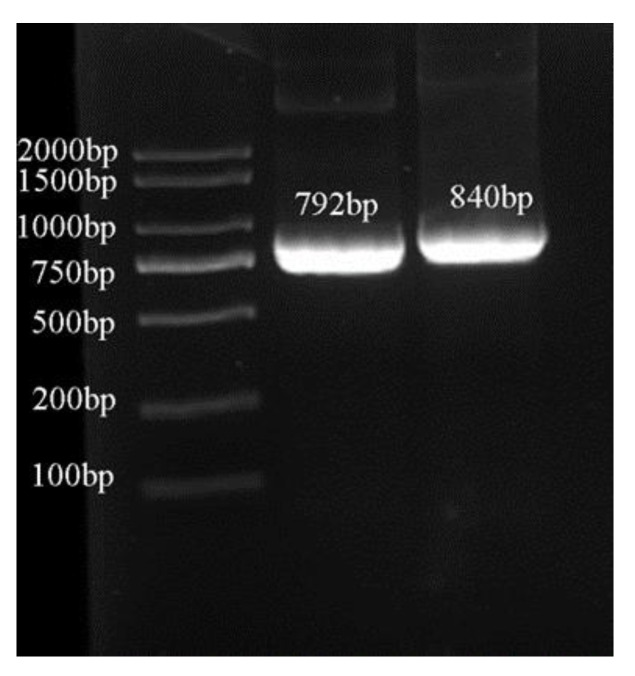
The PCR amplification products of the ORF sequences of *HbOT1* and *HbOT2*. Marker DL2000.

**Figure 2 cells-11-03793-f002:**
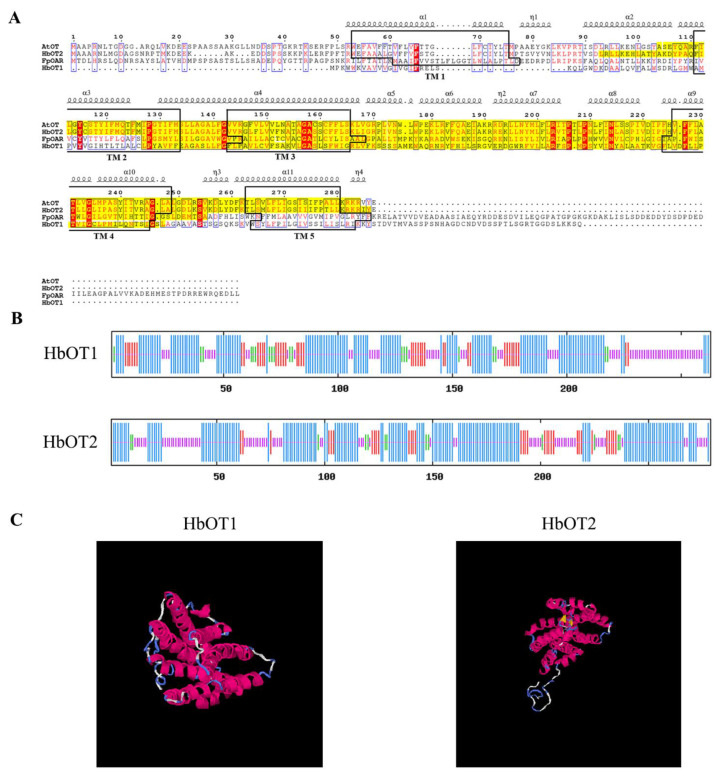
Multiple sequence alignment and protein structure analysis of HbOT1 and HbOT2. (**A**) Multiple sequence alignment of FpOAR, HbOT1, HbOT2, and AtOT. Different secondary structures are labeled above a specific sequence, the black boxes represent the transmembrane domains (TM1-TM5) of the protein, and the yellow highlighted fragment represents the SNARE_assoc conserved domain of the protein. Conserved and similar residues are identified by red shadows and blue boxes, respectively. (**B**) Prediction for the secondary structure of HbOT1 and HbOT2 by NOS@SOPMA. Each bar represents an amino acid, with blue for α-helix, red for extended strand, green for β-turn, and purple for random coil. (**C**) Prediction for the tertiary structure of HbOT1and HbOT2 by I-TASSER.

**Figure 3 cells-11-03793-f003:**
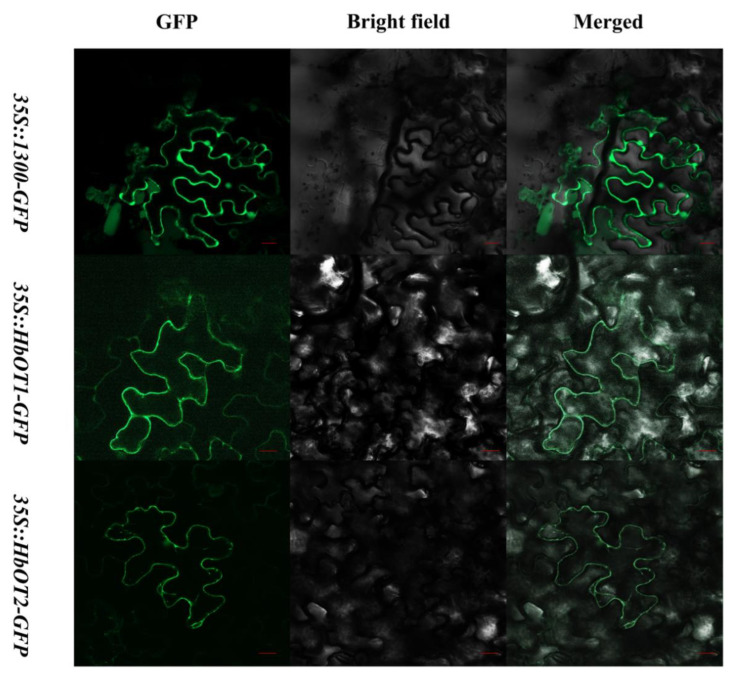
Subcellular localization of *HbOT1* and *HbOT2*. The *35S::HbOT1-GFP* and *35S::HbOT2-GFP* fusion proteins were transiently expressed in tobacco (*N. benthamiana*) leaf epidermal cells. Bars indicate the length of 20 um.

**Figure 4 cells-11-03793-f004:**
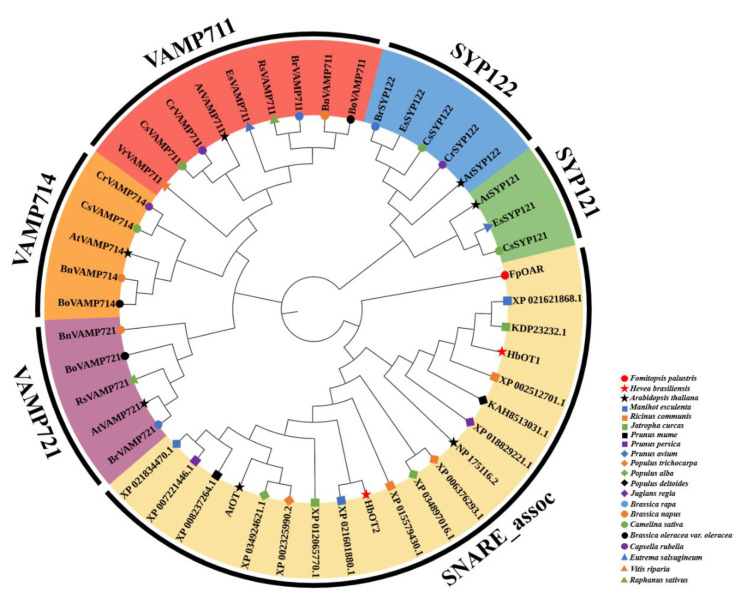
Phylogenetic analysis of FpOAR, HbOT1, and HbOT2 and SNARE protein members in other plants. All amino acid sequences were obtained from the database of NCBI (https://www.ncbi.nlm.nih.gov/) accessed on 7 November 2020, with the GenBank accession numbers indicated. The complete protein sequences were aligned using MEGA 7, and the phylogenetic tree was constructed using the neighbor-joining method and displayed using iTOL (https://itol.embl.de/).

**Figure 5 cells-11-03793-f005:**
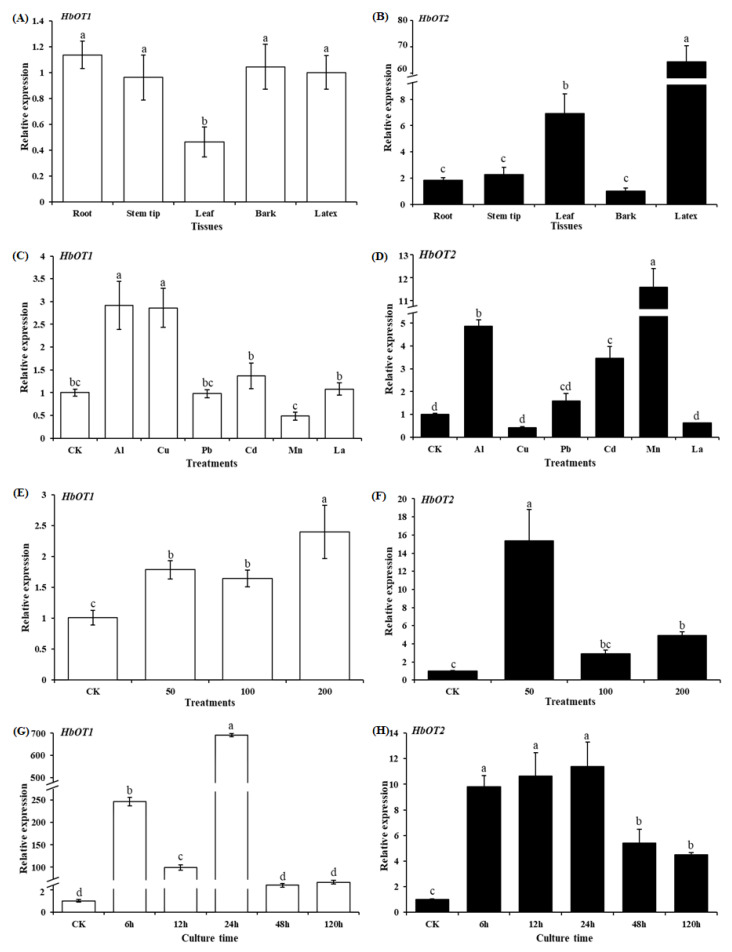
Analysis of *HbOT1* and *HbOT2* expression patterns. (**A**,**B**) qRT-PCR analysis of the *HbOT1* and *HbOT2* transcripts in different tissues of “Reyan 7-33-97” rubber tree saplings. (**C**,**D**) qRT-PCR analysis of *HbOT1* and *HbOT2* under different metal ion stresses. The two-whorled leaf tissue-cultural rubber tree saplings were hydroponically treated with 200 mmol/L AlCl_3_·6H_2_O, CuSO_4_·5H_2_O, PbCl_2_, CdCl_2_, MnSO_4_·H_2_O, and LaCl_3_·7H_2_O, respectively for 120 h. (**E**,**F**) qRT-PCR analysis of *HbOT1* and *HbOT2* in the root of “Reyan 7-33-97” rubber tree saplings under different concentrations of aluminum stresses. (**G**,**H**) qRT-PCR analysis of *HbOT1* and *HbOT2* in the root of “Reyan 7-33-97” rubber tree saplings under aluminum stresses for different time durations. The relative expression values of *HbOT1* and *HbOT2* are given as means ± SE (*n* = 3). The experiments were performed with at least three independent biological replicates. Different letters indicate significant difference among treatments at *p* < 0.05.

**Figure 6 cells-11-03793-f006:**
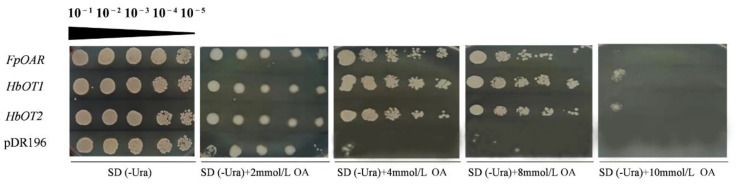
Growth of yeast transformants under different concentration of oxalic acid. The yeast transformants containing pDR196 empty vector and *FpOAR* were used as the negative control and positive control, respectively. The oxalic acid concentrations used were 0 (CK), 2, 4, 8, and 10 mmol/L according to [34] and our series of experiments. The yeast concentrations for each treatment were 10^−1^, 10^−2^, 10^−3^, 10^−4^, and 10^−5^, from left to right.

**Figure 7 cells-11-03793-f007:**
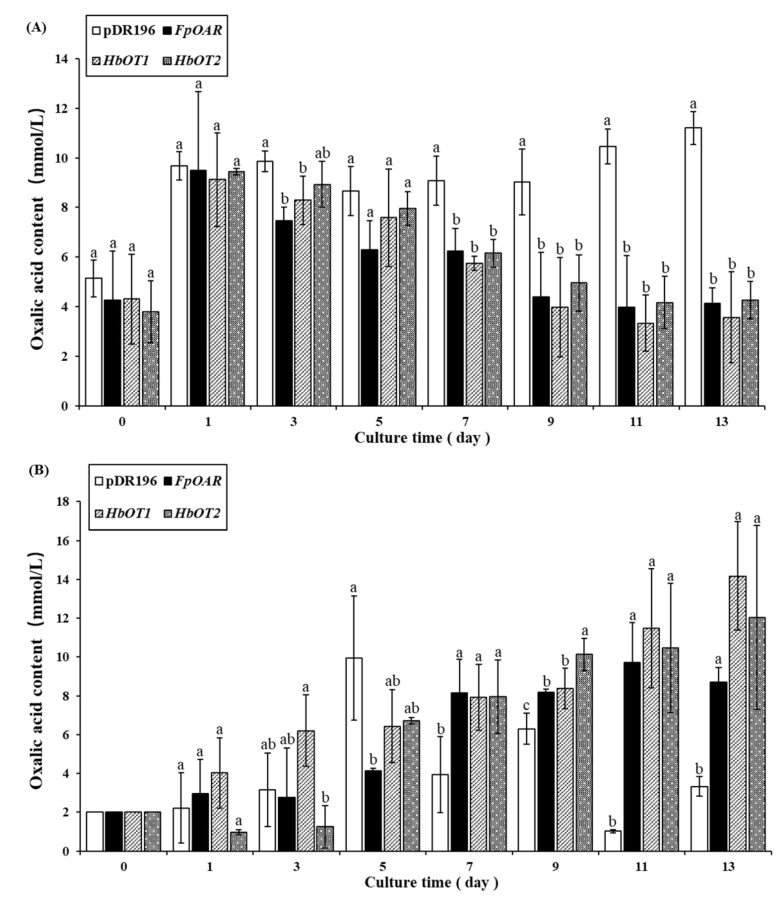
Changes of oxalic acid contents in medium and cells of yeast transformants under 2 mmol/L oxalic acid stress. (**A**) Oxalic acid content in yeast cells. (**B**) Oxalic acid content in the culture medium. Different letters indicate significant difference among different treatments at *p* < 0.05.

**Figure 8 cells-11-03793-f008:**
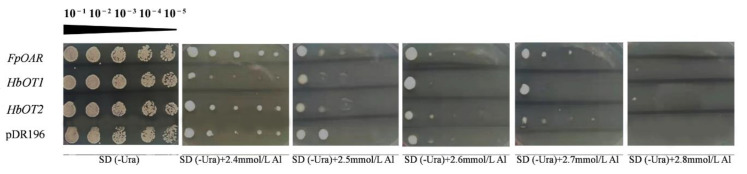
Growth of yeast transformants containing pDR196 empty vector (negative control), *FpOAR*, *HbOT1*, and *HbOT2* on SD (-Ura) plates with 0 (CK), 2.4, 2.5, 2.6, 2.7, and 2.8 mmol/L AlCl_3_. The yeast concentrations for each treatment were 10^−1^, 10^−2^, 10^−3^, 10^−4^, and 10^−5^, from left to right.

**Figure 9 cells-11-03793-f009:**
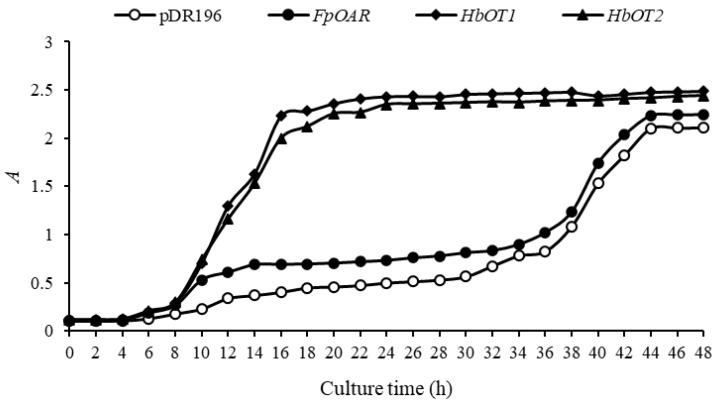
The growth curve of different yeast transformants under 2.7 mmol/L Al^3+^ stress.

**Figure 10 cells-11-03793-f010:**
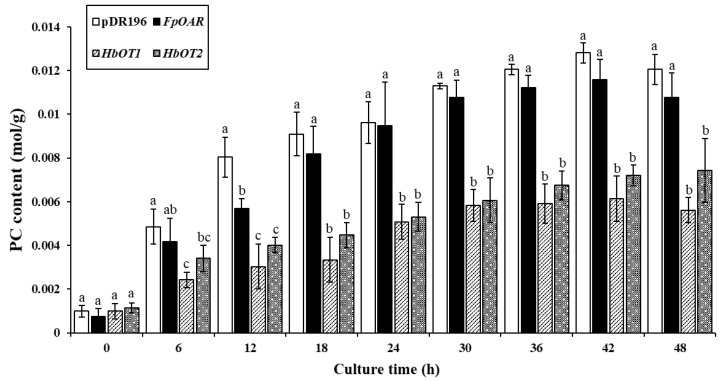
Changes of protein carbonyl content in yeast transformants under 2.7 mmol/L Al^3+^ stress. Different letters indicate significant differences among different treatments at *p* < 0.05.

**Table 1 cells-11-03793-t001:** Primers used in the experiments.

Primer Name	Primer Sequence	Expected Size of PCR Products	Description
*HbOT1*	F: ATGCCGAAATGGTGGAAGGT	792 bp	For ORF sequence cloning
	R: TTATTGACTCTTCTTCAGGCTGTCAC		
*HbOT2*	F: ATGGCCGCGGCGAGGAATCTG	840 bp	
	R: TCAAAAGGAAACCGAAGTACCA		
*HbOT1* (pDR196)	F: TTGGGTACCGGGCCCCCCCTCGAGGATGCCGAAATGGTGGAAGGT	841 bp	For yeast heterologous expression vector construction
	R: CTAGTGGATCCCCCGGGCTGCAGGTTATTGACTCTTCTTCAGGC		
*HbOT2* (pDR196)	F: TGGGTACCGGGCCCCCCCTCGAGGATGGCCGCGGCGAGGAATCT	888 bp	
	R: CTAGTGGATCCCCCGGGCTGCAGGTTACTCATATATCCGCTTTC		
*FpOAR* (pDR196)	F: GGTACCGGGCCCCCCCTCGAGGATGACCGACCTGCATCGAAG	1219 bp	
	R: CTAGTGGATCCCCCGGGCTGCAGGTCAGAGAAGATCTTCTTGCC		
*HbOT1* (pCAMBIA1302)	F: GATCGAATTCCAATGCCGAAATGGTGGAAGGT	814 bp	For subcellular localization vector construction
	R: GATCAAGCTTTTATTGACTCTTCTTCAGGCTGTCAC		
*HbOT2* (pCAMBIA1300)	F: GAGAACACGGGGGACTATGGCCGCGGCGAGGAAT	878 bp	
	R: CAGCTCCTCGCCCTTGCTCACCATGCTCATATATCCGCTTTCTTT		
q*HbOT1*	F: TGGTTGTCTGCCCATGATCT	199 bp	For qRT-PCR
	R: GCTAGGAGAGGATGCAACCA		
q*HbOT2*	F: GGCTGATCATCACCTTCCCT	186 bp	
	R: TCCTAGGAGATTGATTTCTGGCT		
q*HbUBC4*	F: TCCTTATGAGGGCGGAGTC	82 bp	
	R: CAAGAACCGCACTTGAGGAG		

**Table 2 cells-11-03793-t002:** **The** pH variation of culture solution for yeast transformants containing pDR196 empty vector (negative control), *FpOAR* (positive control), *HbOT1*, and *HbOT2* under 13 days’ 2 mmol/L oxalic acid stresses.

Yeast Cell	Culture Time (day)							
	0	1	3	5	7	9	11	13
pDR196	5	4.59 ± 0.082 Aa	3.87 ± 0.095 Ba	3.44 ± 0.125 Ba	3.1 ± 0.066 Ca	2.89 ± 0.118 Da	2.78 ± 0.046 Da	2.76 ± 0.072 Da
*FpOAR*	5	4.09 ± 0.075 Ab	3.19 ± 0.075 Bab	2.76 ± 0.095 Cb	2.64 ± 0.03 Db	2.55 ± 0.053 DEb	2.5 ± 0.066 EFb	2.42 ± 0.05 Fb
*HbOT1*	5	4.12 ± 0.087 Ab	3.13 ± 0.131 Bb	2.67 ± 0.046 Cb	2.49 ± 0.085 Dc	2.4 ± 0.092 DEb	2.32 ± 0.082 Ec	2.31 ± 0.075 Eb
*HbOT2*	5	4.13 ± 0.108 Ab	3.16 ± 0.07 Bb	2.72 ± 0.079 Cb	2.54 ± 0.07 Dbc	2.43 ± 0.066 DEb	2.4 ± 0.066 DEbc	2.36 ± 0.075 Eb

Note: Different uppercase letters show significant differences in treatment time, and different lowercase letters imply significant differences in yeast transformants (*p* < 0.05), the same in the following tables.

**Table 3 cells-11-03793-t003:** Variation of dry weight for yeast transformants containing pDR196 empty vector (negative control), *FpOAR* (positive control), *HbOT1*, and *HbOT2* under 13 days’ 2 mmol/L oxalic acid stress. (g).

Yeast Cell	Culture Time (day)							
	0	1	3	5	7	9	11	13
pDR196	0	0.025 ± 0.01 Ac	0.05 ± 0.01 Ab	0.125 ± 0.018 Bc	0.175 ± 0.015 Cb	0.175 ± 0.02 Cb	0.2 ± 0.023 Cb	0.2 ± 0.015 Cc
*FpOAR*	0	0.05 ± 0.015 Ab	0.125 ± 0.02 Ba	0.175 ± 0.015 Cb	0.225 ± 0.018 Da	0.225 ± 0.018 Da	0.225 ± 0.018 Dab	0.25 ± 0.01 Db
*HbOT1*	0	0.075 ± 0.013 Aa	0.15 ± 0.018 Ba	0.2 ± 0.013 Cab	0.225 ± 0.023 CDa	0.25 ± 0.015 Da	0.25 ± 0.025 Da	0.325 ± 0.013 Ea
*HbOT2*	0	0.05 ± 0.01 Ab	0.15 ± 0.013 Ba	0.225 ± 0.015 Ca	0.225 ± 0.018 Ca	0.25 ± 0.02 Ca	0.25 ± 0.018 Ca	0.325 ± 0.015 Da

## Data Availability

All data are available on reasonable request to the corresponding authors.

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
