# Peer review of "Cloning, Expression Analysis, and Functional Characterization of Candidate Oxalate Transporter Genes of HbOT1 and HbOT2 from Rubber Tree (Hevea brasiliensis)"

_cells, 2022, doi:10.3390/cells11233793_

Round 1
Reviewer 1 Report
The manuscript entitled „Cloning, expression analysis and functional characterization of candidate oxalate transporter genes of HbOT1 and HbOT2 from rubber tree (Hevea brasiliensis)” by Zongming Yang, Pingjuan Zhao, Wentao Peng, Zifan Liu, Guishui Xie, Xiaowei Ma and Feng An tackles with the problem that is important from both scientific and economical point of view. The contamination of soils with metals poses a serious threat to human and animal health and life and plant-based food is an important source of metal in the food chain. Therefore, I found the manuscript timely and significant.
Specific comments:
1. Manuscript needs significant improvement in terms of language. There is several awkward and unclear statements and expressions. To list only a few examples:
- page 2, line 51 “…induction of root pH barier…”
- page 2, lines 60-61 “…and achieving the purpose of aluminum detoxification …”
- page 4, lines 159-160 “….the PCR products were recovered and connected to the pMD-18T vector.”
- page 6, line 252 “The results showed that the possibility of performing the function of signal peptide…”
- page 11, line 362 “The expressing values are expressed….”
- page 12, line 381 “…the positive control formed three white small dot colonies which were indiscoverable,..,”
- page 15, line 464 “…HbOT2 had certain aluminum tolerance, and there was a concentration effect.”
2. Page 3, line 102. It is not clear what Authors means by “..in the early stage..”.
3. Page 3, line 126. Please add the type of metal salts used (here and everywhere in the manuscript where needed).
4. Page 3, lines 136-137. The RNA concentration and purity cannot be detected, rather analysed or examined.
5. Page 3, lines 141-142. It is completely not clear how Authors analysed the purity and concentration of cDNA via gel electrophoresis.
6. Page 3, lines 143-144. It is not clear what Authors means by fluorescent quantitative primers? Primers for qPCR are just primers.
7. Page 3, line 145. Please add the name of the producer of real time PCR machine.
8. Page 3, line 147. Please change “system” to reaction mixture.
9. Page 4, line 149. Please check the concentration of primers. It for sure was not 10 mol/L.
10. Page 4, table 1. It is not clear what Authors means by homologous recombination primers? Homologous recombination was used to construct vectors? Why some fragments of primers described as homologous recombination primers are underlined? It would be also good to add the expected sizes of PCR products.
11. Page 4, lines 155-156. Actually the primers given in Table 1 allow to amplify the coding region (open reading frame) of HbOT1 and HbOT2. Full length is a term used for the full cDNA i.e. coding region and 5’ and 3’ UTRs.
12. Page 4, line 173. Why two different pCAMBIA vectors were used?
13. Page 4, line 176. What method was used to transform Agrobacterium?
14. Page 4, line 180. It is not clear what Authors means by saying that the suspension of Agrobacterium was injected.
15. Page 5, lines 188-191. It is not clear how the vectors were constructed. Using homologous recombination?
16. Page 5, line 225. If protein sequence was used as a query and the proteins from rubber tree were identified then not the genome database was screened.
17. Page 5, line 228. Please change genes to mRNAs.
18. Page 5, line 228-230. Please rewrite this sentence since now it is completely unclear.
19. Page 6, lines 231-237. Based on the results present in Figure 1 Authors did not amplify in PCR the full length cDNAs, just open reding frames. So, in fact the results did not show the full length cDNA.
20. Page 6, lines 242, 245. It is very unusual and not very informative to show the elemental composition of the protein.
21. Page 7, figure 2. Please improve the quality of the figure.
22. Page 12, figure 6. Please include more details in the figure caption.
23. Pages 12-13, table 2 and 3. Please include more details to the table captions.
24. Pages 16-17, lines 514-529, figures 11 and 12. Why those results are presented in discussion? Please move it to results and if Authors do no find them necessary for the main manuscript those results should be included as supplementary material.
25. Page 19, line 601-602. Neither here nor anywhere else in the manuscript the nature of yeast AD108 mutant is properly described.
Author Response
Dear reviewer 1,
Thank you very much for the valuable comments on our manuscript 978409 “Cloning, expression analysis and functional characterization of candidate oxalate transporter genes of HbOT1 and HbOT2 from rubber tree (Hevea brasiliensis)”. We have carefully revised the paper according to the reviewers’ comments. Those comments help us a lot to improve the manuscript. A point by point response to reviewers is attached at the end of this letter.
We think the paper can be now published in Cells. We are looking forward for your further evaluation.
Kind regards
Feng
Reviewer 1
Comments and Suggestions for Authors
The manuscript entitled “Cloning, expression analysis and functional characterization of candidate oxalate transporter genes of HbOT1 and HbOT2 from rubber tree (Hevea brasiliensis)” by Zongming Yang, Pingjuan Zhao, Wentao Peng, Zifan Liu, Guishui Xie, Xiaowei Ma and Feng An tackles with the problem that is important from both scientific and economical point of view. The contamination of soils with metals poses a serious threat to human and animal health and life and plant-based food is an important source of metal in the food chain. Therefore, I found the manuscript timely and significant.
Specific comments:
- Manuscript needs significant improvement in terms of language. There is several awkward and unclear statements and expressions. To list only a few examples:
- page 2, line 51 “…induction of root pH barier…”
Thanks for your suggestion. We have replaced “…induction of root pH barier…” with “…formation of a rhizosphere pH barrier…”.
- page 2, lines 60-61 “…and achieving the purpose of aluminum detoxification …”
Thanks for your reminder. We have deleted “and achieving the purpose of aluminum detoxification”
- page 4, lines 159-160 “….the PCR products were recovered and connected to the pMD-18T vector.”
Thanks for your suggestion. We have replaced “….the PCR products were recovered and connected to the pMD-18T vector.” with “….the target gene fragment was purified and then connected to the pMD-18T vector..”.
- page 6, line 252 “The results showed that the possibility of performing the function of signal peptide…”
Thanks for your reminder. We have replaced “The results showed that the possibility of performing the function of signal peptide…” with “The results showed that the possibility of HbOT1 and HbOT2 to be signal peptides were 0.0376 and 0.0003, respectively, which were less than the threshold of 0.5000.”
- page 11, line 362 “The expressing values are expressed….”
Thanks for your suggestion. We have replaced “The expressing values are expressed….” with “The relative expression values of HbOT1 and HbOT2 are given as….”.
- page 12, line 381 “…the positive control formed three white small dot colonies which were indiscoverable…”
Thanks for your reminder. We have replaced “…the positive control formed three white small dot colonies which were indiscoverable…” with “…the positive control formed three white small dot colonies which were too tiny to be discovered…”.
- page 15, line 464 “…HbOT2 had certain aluminum tolerance, and there was a concentration effect.”
Thanks for your suggestion. We have replaced “…HbOT2 had certain aluminum tolerance, and there was a concentration effect.” with “The results showed that the yeast transformed with HbOT1 and HbOT2 showed stronger aluminum tolerance than the negative control”.
- Page 3, line 102. It is not clear what Authors means by “..in the early stage..”.
Thanks for the valuable comments. We have deleted that.
- Page 3, line 126. Please add the type of metal salts used (here and everywhere in the manuscript where needed).
Thanks for the valuable comments. We have revised that in the text and Figure 5 notes.
- Page 3, lines 136-137. The RNA concentration and purity cannot be detected, rather analysed or examined.
Thank you for the reminder. We have modified that in the manuscript.
- Page 3, lines 141-142. It is completely not clear how Authors analysed the purity and concentration of cDNA via gel electrophoresis.
Sorry for the confusion expression. we preliminary judged the quality of cDNA by observing the brightness and tailing of the target band in the 1% agarose gel electrophoresis. Then, the quality and concentration of cDNA were analysed by measuring the OD with a Nanodrop analyzer and calculating the OD260/280 value.Since these are basic preocess for cDNA and RNA verification, we have deleted these according the other reviewer’s recommendation.
- Page 3, lines 143-144. It is not clear what Authors means by fluorescent quantitative primers? Primers for qPCR are just primers.
Thank you for the reminder. We have corrected the description of primers for qRT-PCR.
- Page 3, line 145. Please add the name of the producer of real time PCR machine.
Thanks. We have added the producer in the manuscript.
- Page 3, line 147. Please change “system” to reaction mixture.
Thanks, we have modified that.
- Page 4, line 149. Please check the concentration of primers. It for sure was not 10 mol/L.
Thanks for your reminder. We have corrected “10 mol/L” to “10 umol/L”.
- Page 4, table 1. It is not clear what Authors means by homologous recombination primers? Homologous recombination was used to construct vectors? Why some fragments of primers described as homologous recombination primers are underlined? It would be also good to add the expected sizes of PCR products.
Thank you for the valuable comments. We have modified that in the manuscript..
- Page 4, lines 155-156. Actually the primers given in Table 1 allow to amplify the coding region (open reading frame) of HbOT1 and HbOT2. Full length is a term used for the full cDNA i.e. coding region and 5’ and 3’ UTRs.
Thank you for the reminder We have modified that in the manuscript.
- Page 4, line 173. Why two different pCAMBIA vectors were used?
Thanks for your reminder. In order to distinguish HbOT1 and HbOT2 transgenic plants in subsequent experiments, we used two plant expression vectors. Since there was very little modification for the vector, we ensured that there was no difference between pCAMBIA1300 and pCAMBIA1302 vectors for the sub-cellular localization experiments.
- Page 4, line 176. What method was used to transform Agrobacterium?
Thank you for the valuable comments. The constructed 35S::HbOT1-GFP and 35S::HbOT2-GFP vector were introduced into the Agrobacterium tumefaciens strain GV3101 by the heat shock method, and we have added the information in the manuscript.
- Page 4, line 180. It is not clear what Authors means by saying that the suspension of Agrobacterium was injected.
Thanks for your reminder. We have modified that in the manuscript.
- Page 5, lines 188-191. It is not clear how the vectors were constructed. Using homologous recombination?
Yes, the vector was constructed using the homologous recombination with a ClonExpress® â…¡ One Step Cloning Kit (Vazyme Biotech Co.,Ltd., Nanjing) according to the manufactures’ manual. We have added the description in the manuscript.
- Page 5, line 225. If protein sequence was used as a query and the proteins from rubber tree were identified then not the genome database was screened.
Thanks for your reminder. We have modified that in the manuscript.
- Page 5, line 228. Please change genes to mRNAs.
Thanks for your reminder. We have modified that in the manuscript.
- Page 5, line 228-230. Please rewrite this sentence since now it is completely unclear.
Thanks for your reminder. We have modified that in the manuscript.
- Page 6, lines 231-237. Based on the results present in Figure 1 Authors did not amplify in PCR the full length cDNAs, just open reading frames. So, in fact the results did not show the full length cDNA.
Thanks for your reminder. We have modified that in the manuscript.
- Page 6, lines 242, 245. It is very unusual and not very informative to show the elemental composition of the protein.
Thanks for the valuable comments. We have deleted that.
- Page 7, figure 2. Please improve the quality of the figure.
Thanks for your reminder. We have modified that in the manuscript.
- Page 12, figure 6. Please include more details in the figure caption.
Thanks for your reminder. We have modified that in the manuscript.
- Pages 12-13, table 2 and 3. Please include more details to the table captions.
Thanks for your reminder. We have modified that in the manuscript.
- Pages 16-17, lines 514-529, figures 11 and 12. Why those results are presented in discussion? Please move it to results and if Authors do no find them necessary for the main manuscript those results should be included as supplementary material.
Thanks for your reminder. This part was just used to show the phenotype change and secretion oxalic acid of rubber tree sapling to aluminum stress, which is the background of this research. These results have been reported in our earlier papers (reference31-33) and are not focus of this manuscript. Therefore, we have re-written this part and removed figures 11 and 12 from the paper.
- Page 19, line 601-602. Neither here nor anywhere else in the manuscript the nature of yeast AD1-8 mutant is properly described.
Thanks for your reminder. The main characteristic of yeast mutant AD1-8 were introduced in the last paragraph of introduction. Yeast mutant AD1-8 is primarily used for identification of transporter protein. It was also used to characterization of oxalic acid transporter FpOAR in fungus in Reference 34. More information about it can be found in reffernce 34 and 54. We have made some modification for the introduction of AD1-8 in the manuscript.
Reviewer 2 Report
Manuscript entitled “Cloning, expression analysis and functional characterization of 2 candidate oxalate transporter genes of HbOT1 and HbOT2 from 3 rubber tree (Hevea brasiliensis)” submitted buy Yang et al, is of significance in the specific field of research. Work has been performed well and presented in accurate way. However, manuscript still needs improvement before the acceptance for publication.
The points are mentioned below
Introduction
Line 36-40: Needs supporting reference
Introduction lacks background information regarding HbOT1 and HbOT2 genes
Introduction should be revised with the hypothesis of the study
Methodology
Section has been written much expanded, it should be reduced at least 20%
Results
Results section needs English and grammar improvement, additionally typing error should be checked like Line 231 ‘cleat” should be “clear”. Additionally this section should also shortened.
Section 3.5 to 3.8 caption should be modified
Fig 5 B and H: I have query regarding letter of significance, why leaf tissues and root and stem tip expression is not significantly differ in figure 5 B. Similarly justify figure 5 H.
Discussion
Section has been written well and supplemented with proper references.
Manuscript can be accepted after implementing above points
Author Response
Dear reviewer 2,
Thank you very much for the valuable comments on our manuscript 978409 “Cloning, expression analysis and functional characterization of candidate oxalate transporter genes of HbOT1 and HbOT2 from rubber tree (Hevea brasiliensis)”. We have carefully revised the paper according to the reviewers’ comments. Those comments help us a lot to improve the manuscript. A point by point response to reviewers is attached at the end of this letter.
We think the paper can be now published in Cells. We are looking forward for your further evaluation.
Kind regards
Feng
Reviewer 2
Comments and Suggestions for Authors
Manuscript entitled “Cloning, expression analysis and functional characterization of candidate oxalate transporter genes of HbOT1 and HbOT2 from rubber tree (Hevea brasiliensis)” submitted buy Yang et al, is of significance in the specific field of research. Work has been performed well and presented in accurate way. However, manuscript still needs improvement before the acceptance for publication.
The points are mentioned below
Introduction
Line 36-40: Needs supporting reference
Thanks for your reminder. We have added relevant references.
Introduction lacks background information regarding HbOT1 and HbOT2 genes
Thanks for your reminder. In fact, we have briefly described the discovery of HbOT1 and HbOT2 by NCBI BLASTP using the Fomitopsis palustris oxalate transporter FpOAR as the query sequence. Since these two genes and its homologous gene AtOT had not any functional annotation in NCBI, we hypothesised that HbOT1 and HbOT2 might have the function of transport oxalic acid as FpOAR. Function characterization of HbOT1 and HbOT2 as oxalic acid transporters is our objective and is the novelty of the paper since none oxalic acid transporter was reported in plants till now.
Introduction should be revised with the hypothesis of the study
Thanks for your suggestion. At present, the research on plant oxalate transporters is still in its infancy, and the research background for reference is very rare. Therefore, in the Introduction, we mainly focused on the importance of root efflux of oxalic acid in aluminum detoxification. The research is problem-driven research to prove that HbOT1 and HbOT2 are oxalic acid transporters as its previously characterized homologous FpOAR. We believe the introduction is more logical in its current form. In case the reviewer is not agree with that, we are happy to and a hypothesis for the study.
Methodology
Section has been written much expanded, it should be reduced at least 20%
Thanks for your reminder. We tried to introduce the experiments very clear. We have deleted some basic and unnecessary information in the methodology in the manuscript.
Results
Results section needs English and grammar improvement, additionally typing error should be checked like Line 231 ‘cleat” should be “clear”. Additionally, this section should also shorten.
Thanks for your reminder. We have modified that in the manuscript. Also, we have used a professional proof-reading company to edit the English.
Section 3.5 to 3.8 caption should be modified
Thanks for your reminder. We have modified that in the manuscript.
Fig 5 B and H: I have query regarding letter of significance, why leaf tissues and root and stem tip expression is not significantly differ in figure 5 B. Similarly justify figure 5 H.
Thanks for the valuable comments. By examining the raw data, we found the error for the Figure 5 B significance mark was wrong as you said. There were significant differences between the expression of leaf tissue and the root and stem tissues. We have corrected the figure mark and discriptions in the manuscript. In addition, we checked all expression analysis data and didn’t find any errors for other figures.
Discussion
Section has been written well and supplemented with proper references.
Thanks for the positive feedback.
Manuscript can be accepted after implementing above points
Thanks for the positive recommendation.
Round 2
Reviewer 1 Report
The manuscript has been substantially improved. Before publication please only small issues need to be addressed.
1. Please replace “u” with “μ” throughout the manuscript.
2. The concentration of primers is not 10 umol/L but rather 10 μmol/μL.
3. Figure 1 – I think it is too large.
4. Line 253 – please remove “in our research”.
Author Response
Dear reviewer1,
Thank you very much for the valuable comments on our manuscript 978409 “Cloning, expression analysis and functional characterization of candidate oxalate transporter genes of HbOT1 and HbOT2 from rubber tree (Hevea brasiliensis)”. We have carefully revised the paper according to the reviewers’ comments. Those comments help us a lot to improve the manuscript. A point by point response to reviewers is attached at the end of this letter.
We think the paper can be now published in Cells. We are looking forward for your further evaluation.
Reviewer 1
Comments and Suggestions for Authors
The manuscript has been substantially improved. Before publication please only small issues need to be addressed.
Thanks for the positive feedback.
- Please replace “u” with “μ” throughout the manuscript.
Thank you for the reminder. We have modified that in the manuscript.
- The concentration of primers is not 10 umol/L but rather 10 μmol/μL.
Thanks, we have modified that.
- Figure 1 – I think it is too large.
Thank you for the reminder. We have adjusted the picture size to be smaller.
- Line 253 – please remove “in our research”.
Thank you for the reminder. We have deleted that.